# Bayesian Networks to Support Decision-Making for Immune-Checkpoint Blockade in Recurrent/Metastatic (R/M) Head and Neck Squamous Cell Carcinoma (HNSCC)

**DOI:** 10.3390/cancers13235890

**Published:** 2021-11-23

**Authors:** Marius Huehn, Jan Gaebel, Alexander Oeser, Andreas Dietz, Thomas Neumuth, Gunnar Wichmann, Matthaeus Stoehr

**Affiliations:** 1Head and Neck Surgery, Department of Otorhinolaryngology, University Hospital Leipzig, 04103 Leipzig, Germany; marius.huehn@medizin.uni-leipzig.de (M.H.); Andreas.Dietz@medizin.uni-leipzig.de (A.D.); Gunnar.Wichmann@medizin.uni-leipzig.de (G.W.); 2Innovation Center Computer Assisted Surgery (ICCAS), Faculty of Medicine, University Leipzig, 04103 Leipzig, Germany; Jan.Gaebel@medizin.uni-leipzig.de (J.G.); Alexander.Oeser@medizin.uni-leipzig.de (A.O.); Thomas.Neumuth@medizin.uni-leipzig.de (T.N.)

**Keywords:** head and neck squamous cell carcinoma (HNSCC), immunotherapy, immune checkpoint blockade (ICB) targeted therapy, Bayesian network, molecular tumor board, multidisciplinary tumor board, clinical decision support system (CDSS)

## Abstract

**Simple Summary:**

Tumor therapy in many human malignancies, including head and neck cancer, is increasingly demanding due to advances in diagnostics and individualized treatments. Multidisciplinary tumor boards, especially molecular tumor boards, consider a great amount of information to find the optimal treatment decision. Clinical decision support systems can help in optimizing this complex decision-making process. We designed a digital patient model based on conditional probability algorithms as Bayesian networks to support the decision-making process regarding treatment with approved immunotherapeutic agents (Nivolumab and Pembrolizumab). The model is able to process relevant clinical information to recommend a certain immunotherapeutic agent based on literature, approval, and guidelines.

**Abstract:**

New diagnostic methods and novel therapeutic agents spawn additional and heterogeneous information, leading to an increasingly complex decision-making process for optimal treatment of cancer. A great amount of information is collected in organ-specific multidisciplinary tumor boards (MDTBs). By considering the patient’s tumor properties, molecular pathological test results, and comorbidities, the MDTB has to consent an evidence-based treatment decision. Immunotherapies are increasingly important in today’s cancer treatment, resulting in detailed information that influences the decision-making process. Clinical decision support systems can facilitate a better understanding via processing of multiple datasets of oncological cases and molecular genetic information, potentially fostering transparency and comprehensibility of available information, eventually leading to an optimum treatment decision for the individual patient. We constructed a digital patient model based on Bayesian networks to combine the relevant patient-specific and molecular data with depended probabilities derived from pertinent studies and clinical guidelines to calculate treatment decisions in head and neck squamous cell carcinoma (HNSCC). In a validation analysis, the model can provide guidance within the growing subject of immunotherapy in HNSCC and, based on its ability to calculate reliable probabilities, facilitates estimation of suitable therapy options. We compared actual treatment decisions of 25 patients with the calculated recommendations of our model and found significant concordance (Cohen’s *κ* = 0.505, *p* = 0.009) and 84% accuracy.

## 1. Introduction

Head and neck squamous cell carcinoma (HNSCC) is one of the most frequent solid cancers with about 38,000 new cases in the US and 10,860 estimated deaths annually [1]. In Germany, an incidence of 14,424 new cases and 6028 deaths associated with HNSCC were registered in 2020. These numbers comprise cancers of the larynx, pharynx, lips, and oral cavity [2]. HNSCC is a highly invasive tumor, and often surgery and radiotherapy are not enough to defeat this cancer, so systemic treatment and chemotherapy agents are necessary [3]. There are many different chemotherapeutic agents applied in head and neck oncology, e.g., platinum derivatives, taxanes, or 5-fluorouracil. These exert an impact on all cells, particularly causing damage to rapidly proliferating cells, both cancer and somatic cells, e.g., in bone marrow or gastrointestinal tissue, potentially causing severe treatment side effects. Today’s medicine can offer more precise so-called targeted drugs, for instance, Cetuximab, as part of immunotherapy. HNSCC is not only highly invasive, but shows huge heterogeneity in tumor-infiltrating immune cells and eventually high immunogenic activity [4]. Systemic treatment with antibodies can trigger effects on tumor cells (over-) expressing designated surface antigens, like the epidermal growth factor receptor (EGFR), programmed cell death protein 1 (PD-1), its ligand (PD-L1), as well as the cytotoxic T lymphocyte-associated protein 4 (CTLA-4). These antigens, as many other alike, can contribute to immunologic synapses. The immunologic synapse is a complex structure or different surface antigens, linking different antigens of two cells, e.g., a T cell and a tumor cell. This connection can either cause immune suppression and thus potentially foster the tumor’s immune escape or cause the opposite: activation of the host’s immune system. By reducing the impact on non-tumor cells, immunotherapeutic agents hold great potential and have shown promising results in the treatment of many tumor entities. Within the recent past, the Food and Drug Administration (FDA) approved three antibodies for the treatment of metastatic or recurrent (R/M) HNSCC, which augment conventional tumor chemotherapy. Immune checkpoint blockade (ICB) leads to an inactivation of the tumor’s immune evasion capabilities by re-enabling the host’s immune cells to target tumor cells [5,6,7].

Nivolumab as well as Pembrolizumab, both targeting PD-1, showed a longer overall survival (OS) in R/M HNSCC compared to conventional chemotherapy [8,9]. Ferris et al. reported a significantly prolonged OS for treatment with Nivolumab compared to a standard therapy of methotrexate, docetaxel, or Cetuximab. The 231 patients receiving Nivolumab showed a median OS of 7.5 months (95% confidence interval (CI) 5.5 to 9.1) versus 5.1 months (95% CI 4.0 to 6.0) of the 130 patients receiving standard therapy [8]. Likewise, Pembrolizumab achieved a median OS of 8.4 months versus 6.9 months in the common treatment group (hazard ratio (HR) 0.8, 0.65–0.98) [9]. As antibodies blocking the interaction of PD-1 expressed on tumor-infiltrating lymphocytes (TILs) including natural killer (NK) and T cells, the effective application of both ICB agents requires further information about the susceptibility of the tumor towards ICB, based on the expression of the targeted antigen PD-1 [10]. Yet a crucial limitation is not merely the low overall response rate to ICB but also the occurrence of immune-related adverse events (AEs) [11,12]. To set proper indications, besides the TNM classification, successful PD-1 blockade may require a specific immunological context and/or tumor mutational status. The TNM classification, with primary tumor extend (T), loco-regional lymph node affection (N), and distant metastasis (M), mainly describes the anatomic extension of cancer within the patient [13]. Biomarkers obtained through histopathological examination supported by immunohistochemistry and molecular investigations led to the description of a subgroup of HNSCC patients, especially in oropharyngeal cancer, characterized by p16 expression and detectable human papillomavirus (HPV) DNA, demonstrating better treatment response and prognosis compared to those not HPV-related, which are primarily linked to environmental exposure and/or noxa-driven mutations [14]. HNSCC shows a variety of mutations and other micro pathological disorders [15]. Several typical alterations are present in HNSCC cells. Mutated NOTCH1/2/3 and the most common mutation in p53 are frequently detected and cause impaired terminal differentiation, mutated PIK3CA, or CASP8 interfere with cell death [16]. As neoplastic transformed cells expressing such somatic mutations are mostly deleted by T cells, mechanisms regulating their immune reactivity might prevent effective immune surveillance. Among others, PD-L1 expressed on tumor cells by binding to PD-1 on TILs impairs their lytic activity but ICB interfering with the tolerogenic PD-1–PD-L1 interaction may restore immune function. Indeed, PD-L1 expression is linked to a greater benefit from Pembrolizumab therapy, as shown in the KEYNOTE-040 trial [9], but a significant benefit was also observed in patients with PD-L1-negative tumors (KEYNOTE-012, KEYNOTE-055) [17,18].

Like PD-1, CTLA-4 balances the immune response, usually stopping an exaggerated immune response, so the immune system does not damage the host. Cancers abuse this mechanism by the secretion of tumor growth factor β (TGF-β), causing CTLA-4 expression on T cells and expression of B-7, a ligand of CTLA-4. This combination leads to an exhaustion of T cells, allowing the cancer to escape the immune response. Another way to bypass the immune system is achieved by the T cell immunoglobulin mucin-3 (TIM-3), which is commonly expressed by cells of the immune system. After binding its ligand, galectin-9, TIM-3 induces apoptosis and exhaustion in NK cells and T cells [19]. Furthermore, it promotes the expansion of bone marrow-derived suppressor cells (MDSCs), preventing an adequate immune response. Anti-TIM-3 antibodies led to decreased counts of MDSCs and increased T cell activity in an HNSCC mouse model [20].

Other alterations may modify the response of immune checkpoint inhibitors: microsatellite instability (MSI) and a high tumor mutational burden (TMB) and the total number of mutations per coding area [21,22] are indirect measures of tumor antigenicity, causing greater susceptibility to immune therapeutics. Aneuploidy, either caused by hypomethylation of histones or other processes linked to a higher amount of copy number variations including loss of heterozygosity, is associated with a reduced susceptibility to immune checkpoint inhibitors [23,24,25]. HNSCC presents various immune escape mechanisms, demonstrating the importance of drug combinations or even multi-targeted drug regimens to improve the potential of immune-checkpoint inhibitors in future clinical trials.

Although therapies for HNSCC treatment have been approved by the FDA, HNSCC still bears many potential options for targeted therapy [26]. Besides the mentioned antigens, other mediators of the immune reaction may be expressed by the tumor cells evading the immune system. For example, the lymphocyte activation gene (LAG-3) is linked to the inhibition of NK cells, altering their immunologic abilities, resulting in a favorable immunologic milieu for tumor cells [26]. NK cell functions are also altered by the T cell immunoglobulin and immune receptor tyrosine-based inhibitor motif (TIGIT), which causes a reduction of proinflammatory cytokines and reduced immunologic response [20,27].

In order to differentiate and understand the genetic alterations of this heterogeneous group of cancers, gene sequencing approaches and in particular next-generation sequencing of druggable mutations are useful to identify and treat them appropriately. Among others, Tafe et al. [28] reported about different-sized gene panels to identify the most relevant coding mutations, which are useful for the creation of data for supporting informed tumor board decisions.

Despite many genetic alterations, various other factors can influence tumor genesis and the therapeutic process. The immune system shifts towards a misbalance in older patients [29]. The thymus releases a greater amount of regulatory T cells, causing a weakened immune response. Simultaneously, antigens like TIM-3 or PD-1 are upregulated on T cell surfaces, leading again to a reduced immune response [30,31]. Although the immune response toward the cancer may be reduced, side effects during ICB therapy occur on comparable levels in young and old adults [30].

Beside unalterable factors, the patient’s diet and microbiome can affect immunogenic processes. Diets can alter cytokine levels, affecting the treatment with certain ICB in positive ways [32,33,34,35].

In HNSCC, many of the above-mentioned molecular pathological alterations are observed only in subsets of patients, resulting in the demand for appropriate biomarkers informing evidence-based decision-making in the molecular tumor board. We have to emphasize that today, only few targeted agents are approved for treating HNSCC patients. Consequently, clinical data in the context of molecular characteristics and the value of particular biomarkers for decision-making mainly exist for the approved agents. However, difficulties exist in the interpretation of pathologic reports and the therein included molecular data. This poses a risk for achieving maximum efficiency in their regular use in clinical routine despite the limited number of available treatments approved. Such difficulties might become even more relevant if larger panels from companion diagnostics, hotspot mutation-targeted sequencing panels, or from whole exome sequencing and RNA sequencing are available and have to be interpreted and integrated within a reasonable time, resulting in special demands for molecular tumor boards (MTBs).

In contrast to conventional tumor boards, MTBs represent a different approach in cancer treatment [28,36,37,38]. Currently, physicians mainly discuss treatment options restricted to surgical options for specific anatomical sites, and this is often with reference to medical imaging to assess resectability. In MTB, imaging is less relevant, and the board consists not only of a group of physicians, but scientists and molecular pathway specialists also join their ranks. The attending physician introduces the patient and the patient’s molecular signatures to the MTB by also presenting the scientific knowledge about the available targeted therapies, considering evidence from a prior literature review. Consequently, various treatment options are assessed and either an approved therapy, admission to a clinical trial, or an available off-label therapy targeting a potential driver mutation could be suggested. Nevertheless, these mentioned new technologies and investigations come with a rising volume of information that needs to be processed in order to lead to the best patient-related decision and to be usable within a clinical context [38]. In order to involve many different information entities and to lead to an evidence-based patient-specific decision, clinical decision support systems (CDSSs) can assist the physician during the decision-making process by presenting pre-processed multidimensional information to facilitate evidence-based interdisciplinary discussion.

Cypko and Stoehr [39,40] presented the idea of modeling the clinical causalities regarding solid tumor TNM to support the physician in verifying the discussed TNM staging, before transferring it to the tumor board. In this study, this approach of representing the decision-making process regarding immunotherapy or alternative treatment and therapy options was adopted. This paper shows a formal approach to create a Bayesian network model for clinical treatment decision support, illustrating this according to the current state of molecular therapy options in R/M HNSCC, while respecting anticipated changes in clinical practice guidelines for this entity. Thus, we demonstrate that the Bayesian network model provides an objectified treatment recommendation by considering the patient’s properties and certain tumor characteristics forming the basis for utilization of a Bayesian network model as CDSS for molecular tumor boards.

## 2. Methods

### 2.1. Literature Review

We aimed to include the existing medical knowledge regarding the mutational landscape of HNSCC and its consequences for medical treatment into our model. Starting our search systematically, we searched MEDLINE, accessed through PubMed, Embase, Cochrane Library, and Web of Science databases up to October 2021, regarding the mutational landscape of HNSCC, immunotherapy for HNSCC, and potential adverse effects along with managing them as well as the structure and functionality of the MTB. Our initial search with the query “head and neck squamous cell carcinoma survival” found about 69,000 entries in total. We further searched with the following terms: “HNSCC“ (MeSH Terms) OR “Head And Neck Squamous Cell Carcinoma” OR (“Head and Neck” AND “Neoplasm”) AND “Immunotherapy” AND/OR “survival” OR “tumor board” or “molecular tumor board” OR “Immune Checkpoint Inhibitors” or “PD-1 Inhibitors” AND “Survival” OR “Mutational Landscape” OR “Bayesian Network” or “BN” or “Head And Neck Squamous Cell Carcinoma” AND “prognosis” AND*OR “Immunotherapy”.

Additionally, novel targeted therapies now approved for HNSCC treatment led to changed therapy decision paradigms and prioritized new treatment regimens [37,41,42], e.g., Nivolumab or Pembrolizumab for systemic therapy of patients without curative treatment options. These changes were integrated in international accepted clinical practice guidelines [42,43].

To evaluate indications and usability, we studied German medical guidelines regarding the oral cavity and the larynx as well as the current NCCN guidelines for head and neck cancer [42,43,44].

### 2.2. Bayesian Networks

Bayesian networks (BNs) are mathematical models described by an acyclic directed graph of categorical variables and their depending probabilities [39]. The variables contain a set of different states that depict the possible manifestations. Edges represent the direct dependencies of two variables and connect them, resulting in a causality that is characterized by a conditional probability table (CPT). This CPT represents the stochastic prior relationships used to calculate the probability of one state influencing the occurrence of another. Such a model, depicting a medical use case, can be supplied with findings from clinical observations (e.g., clinical aspects, diagnoses, and/or results from histological examination) from one specific case to instantiate a patient-specific model. A Bayesian inference algorithm will then calculate the likelihood of unobserved (or unobservable) variables, e.g., therapy options for a.ny given cancer type.

### 2.3. Creating the Model

The process of model creation is visualized in Figure 1. Firstly, we started by converting the clinical knowledge and procedures into a graphical structure. In the second step, we entered the values for the CPT. Variables of the graph represent possible clinical observations, like side effects and their management/manageability, TNM, or (prior) used drugs. Corresponding states can take on different forms, e.g., simple positive/negative or male/female dichotomous variables, T0, T1 … T4b or N0, N1 … N3b categories, or, for instance, a set of other specific events.

Using the modeling software GeNIe (GeNIe Version 2.2, distributed by Bayesfusion, Marek J. Druzdzel et al., University of Pittsburgh, Pittsburgh, PA, USA, https://www.bayesfusion.com/genie/, accessed on 13 March 2018) allowed the entry of additional properties to each node. We added related literature references to the nodes of the model in the form of a Digital Object Identifier (DOI). The use of DOIs allows for tracking of the sources of evidence. Following the physicians’ trains of thought towards a treatment decision, we set nodes whenever a critical decision within the hierarchical decision tree will be made, e.g., treatment with Pembrolizumab, with the possible outcomes as states, e.g., positive or negative, to obtain information from the underlying dependencies, which one expects to have the higher probability. Necessary clinical information to perform these calculations was modeled as parental nodes, influencing the decision to be made, like PD-L1 status or the single TNM nodes.

Separate nodes represent medical items or entities (e.g., nodes for the TNM categories for staging). Their respective manifestations were included as categorical states. Other clinical matters were summarized and represented as one node with consolidated states (e.g., node for radio-sensitivity or radio compatibility). Several thematically related groups make up the whole model, organized as follows.

#### 2.3.1. Therapy Preconditions

Therapy preconditions cover the vital aspects referring to the design of KEYNOTE-012 (NCT01848834). Only patients with ECOG 3 and better were included within the analysis and R/M HNSCC as the central inclusion criterion [45].

A cluster of nodes represents the widely used TNM classification. Sticking to the randomized controlled trials (RCTs) KEYNOTE-012 (NCT01848834) and KEYNOTE-048 (NCT02358031), only R/M HNSCCs were included. Furthermore, the preconditions give insight into certain qualifications, enabling the patient to gain and endure radiation therapy. In addition, the radio-sensitivity node displays findings that PD-L1 might indicate radio sensitivity [46].

#### 2.3.2. Molecular Tumor Information

The molecular tumor information marks the key point of this model. We selected current clinical targets in HNSCC treatment, like PD-1 and EGFR, which are commonly used targets in HNSCC but also included anticipated clinical targets in the form of HRAS [41]. Furthermore, we chose CTLA-4 as less strongly anticipated but yet important [39]. The combined positive score (CPS) and the tumor proportion score (TPS) are immunohistochemical scores within the indication of treatment with Pembrolizumab and partly Nivolumab to estimate PD-L1 expression stained using the antibody DAKO 22C3. The TPS means the percentage of tumor cells, showing PD-L1 staining, relative to all viable tumor cells. The CPS is the proportion of all PD-L1-expressing cells (tumor cells, lymphocytes, macrophages) to the number of all tumor cells. The TPS means the percentage of tumor cells, showing PD-L1 staining, relative to all viable tumor cells.

#### 2.3.3. Therapy Options

Targeted agents, such as Cetuximab, Nivolumab, Durvalumab, Tipifarnib, and Pembrolizumab, as well as systemic chemotherapy or palliative radiation therapy mark potential and/or parallel treatment solutions, depending on the patient and tumor factors.

#### 2.3.4. Annotation of Probabilities

The CPT needs to represent current clinical knowledge to be able to represent clinical decisions, e.g., how likely a disease presentation is whenever a particular symptom is found, the likelihood *P* (x_1_|x_2_). The likelihood can also be used to find out which therapy out of a spectrum of options will probably be the most appropriate for this specific patient. We used clinical guidelines, FDA approvals, and related clinical trials as well as reliable scientific publications as sources to fill the CPT.

The probability values were set with the software GeNIe, representing the probability of certain states, which were between 1% and 99% based on the results of our literature research. Single states or combinations can affect the outcome nearly independently of any other states, e.g., if the performance status is set to ECOG 5, naturally no treatment will follow (ECOG 5 = death), but for other ECOG states, the available treatment options are shown accordingly.

In the case of yet remaining constellations that cannot be predicted by only one determining probability, we needed to set the probabilities of the remaining combinations, evaluating the single states by respecting the patient’s total profile.

### 2.4. Model Verification

According to continuous feedback from board-certified physicians about the structure of the model and the causalities depicted within, we updated the probabilities according to its purpose. We additionally conducted interviews with experts and asked them to review the state of the model. Clinical scientists, experienced physicians of the head and neck oncology department, and computer scientists reviewed the model on a regular basis to ensure a valid network, based on the rules of mathematical requirements and medical correctness while also respecting the anticipated upcoming changes of the current guidelines.

### 2.5. Model Validation

After establishing the model, we performed a retrospective validation study using clinical information from 25 consecutive cases discussed in our multidisciplinary tumor board asking for “R/M HNSCC patients and immunotherapy decision” as the inclusion criteria for case selection. We compared the recommendation proposed by the BN and the consented decision of the tumor board for the treatment that should be applied utilizing *Pearson’s Chi-square* (χ^2^) test and assessed the sensitivity, specificity, and accuracy of the predictions made by using SPSS version 24 (IBM Corp. IBM SPSS Statistics for Windows, Version 24.0, 2016. Armonk, NY, USA: IBM Corp.). A *p*-value < 0.05 was regarded as significant.

## 3. Results

### 3.1. The Molecular Pathological Model

The model we created consists of 28 nodes. Several nodes are grouped, as indicated by the color in Figure 2, representing their thematic proximity.

To describe the decision routine, we demonstrate the processes in the subgroup of Pembrolizumab and Nivolumab. This cluster can be provided with observed or collected information entities.

### 3.2. Application of the Submodel

In our analysis, we entered the data of a potential patient, with the following features (displayed in Figure 3): M-state: M1; T-state T4b; N-state: N2; ECOG: 1; recurrence: positive; progression during or after platinum-based therapy: negative; PD-L1 expression of tumor cells: TPS > 50%; and combined positive score for PD-L1 expression between 1 and 20: CPS > 1. After the interference algorithm calculated the probabilities of the unobserved states, our model presented the chance of a useful Pembrolizumab application of 90%, which coincides with current medical guidelines and KEYNOTE-012. According to available evidence for the use of Pembrolizumab in R/M HNSCC patients with a known PD-L1 status of TPS > 50% and CPS between 1 and 20, Pembrolizumab combined with cisplatin-based chemotherapy should be used. Despite approval for first-line R/M HNSCC, the use of Nivolumab is not covered by the indication criteria for such R/M HNSCC patients with no prior platinum treatment.

### 3.3. Validation of the Submodel

To test the reliability of the decision model, we compared the calculations of the model and the documented treatment. The required information for validation was extracted from a dataset consisting of 25 HNSCC patient cases that were discussed in the head and neck tumor board and treated in the university hospital Leipzig. The primary information, as described in Table 1, was entered into the model and the results calculated by the model were compared with the treatment conducted as described in column “actual therapy” in Table 1. With an odds ratio of 18 (95% CI 1.38–235.69; *p* = 0.00916), the model achieved a sensitivity of 94.7% and specificity of 85.7%, resulting in a Youden index of 0.812 and accuracy of 84.0% in predicting the actual treatment that the patients received. In line with this, Cohen’s Kappa of *κ* = 0.505 indicates significant concordance of the findings (*p* = 0.009).

The false discovery rate of about 16% (four cases) in the model’s calculations is explained by factors not included in the graph. Two patients were enrolled in randomized clinical trials receiving immunotherapy with Nivolumab as the study treatment, while both suffered from R/M HNSCC but neither had previously received any platinum-based therapy; both treatments may be regarded as off-label use. One patient received Pembrolizumab as part of a study, although he suffered from R/M HNSCC and prior platinum-based therapy legitimates the use of Nivolumab. The remaining patient received treatment with Nivolumab. The present R/M HNSCC and the previous platinum-based therapy legitimate the use of Nivolumab, but high CPS values suggest therapy with Pembrolizumab.

## 4. Discussion

In this paper, we present a modeling approach, reproducing a physician’s line of thinking towards the application of immune therapeutics. We created a BN model to represent the likelihood of particular decisions based on medical causalities in a formal way. Although clinical observations are accurately processed within the BN and lead to suitable results, abstracted patient factors and generalized medical guidelines processed by medical treatment experts have to be adopted to the individual case and therefore adjusted by data obtained in the real world. Consequently, the model is a compromise between the complexity of the model and the possibility to assess every possible individual decision with reasonable certainty (or uncertainty, respectively). To some extent, the size of the model grows through the addition of new dependencies between nodes. An excess of edges results in massively growing numbers of CPTs and it might, in the worst case, restrict the computability of the model. Such inflation also exacerbates the process of setting the individual probabilities of single connections between two nodes. Furthermore, missing or undefined information, such as “Nx” or other unknown states, expressing missing knowledge, is effectively impossible to match with a clearly defined probability, since no vested decision can be made upon significant but unclear observational data. Additionally, even the most unlikely event may not be stated with a probability of zero, since BNs tend to become imprecise by setting a state with an absolute value of zero.

We did not include factors like age, diet, or specific colonization of either the gut or oral microbiome in our model. Measured data and potential (treatment-) consequences are not well established, and consequently not measured. The presented functioning model is a submodel of our main model that was concentrated by current clinical and therapeutic relevance. Therefore, we avoided the inclusion of any further information for the reason of traceability of the model. A wider spectrum of different nodes could depict a more exact model of the patient, but only if adequate data is available that could be connected in a comprehensible manner. This includes potential influencing parameters like the patient’s diet, microbiome, or other patient-related factors. Therefore, the model can calculate the probability of treatment with Pembrolizumab or Nivolumab based on the approvals but not on other potentially influencing parameters.

Immunohistochemistry is a vital factor to set a proper indication for the use of Pembrolizumab. The Combined Positive Score (CPS), as used in KEYNOTE-012 and KEYNOTE-048, permits either a combined use with platinum/5-fluorouracil, if CPS > 1, or even the use of Pembrolizumab as a monotherapy if CPS > 20. The prescribing information also refers to a TPS ≥ 50% as a valid indication for single usage. In contrast, according to CheckMate-141 [8], Nivolumab is approved for R/M HNSCC without prior diagnosis and TPS or CPS scoring of PD-L1 expression. Therefore, these differences in approval should be considered in the decision rules and indeed, the different approvals are adequately represented within the model as single nodes for each possible drug, referring to approval for slightly different indications according to available PD-L1 expression scores. Dependent on these, different states of the model show the resulting probabilities for particular treatments facilitating the decision-making process.

Cypko et al. created a model with a focus on TNM, structuring it according to macro and micro pathological aspects, yet without considering immunohistochemical findings [39]. Possible integration into this greater model makes our model a valuable add-on since the previous model is not able to differentiate between other specific treatment options except for immunotherapy in R/M HNSCC.

Nevertheless, immunotherapy marks a promising approach to HNSCC patients, with a growing number of approved drugs, but with the need for even more identified biomarkers and druggable targets in HNSCC. In the future, our approach may help find the best target therapy approved or indicated within a probably growing number of therapeutic agents and potential targets for treatment of HNSCC. Due to the accessible structure, the submodels representing these potentially available new drugs for new targets can be added easily to the existing model. This possibility of updates supports the approach and is strongly recommended regarding the yet unapproved therapeutics and currently not respected alteration by clinical means. 

Probabilistic models are well established within medical decision support, e.g., Leibovici et al. created a model for choosing the most suitable antibiotic, and its use reduced the hospital stay and the use of broad-spectrum antibiotics [47]. Sesen et al. used BN to predict survival and could show that BNs are reliable tools to do so [48].

In comparison, Leibovici et al. presented a more complex model with more nodes and edges, which was able to propose the right antibiotic treatment in up to 85% of 1203 patients in their assessment. In 25 R/M HNSCC cases tested, our BN model correctly predicted 84% of all treatment decisions, proving its operability. We are aware that the evaluation of the model with 25 patients cases implies limited expressiveness. The presented analysis was conducted for the purpose of providing a proof of concept. The results of our validation study are significant despite only 25 patients being analyzed. In addition, study cohorts with the same pathological survey and detailed patient data are not available in an open-source formant to our knowledge. The lower number of nodes and edges depicts a more basic and less comprehensive modeling approach, causing a less detailed graph. The lower count of different treatment options and included studies limits the decision quality, but the lesser internal connections and more direct structure of the model allow an easy expansion with new approved drugs and therapy regimens. With the included literature sources for each node, the model may aid in a literature review and offer guidance within the decision-making process. Thus, the BN could also be used in teaching or training scenarios. 

While the application of antibiotics related to bacterial infection is an objective and fact-based decision-making process, tumor therapy is not. Fifty percent of the deviations from the calculated treatment resulted in best supportive care (BSC), although immunotherapy would have been the treatment of choice. It is impossible to respect subjective and personal decisions within the model, resulting in its inability to include them in its calculations. In our case, BSC is not yet included within the calculating model, but in the cases stated above, it would not cause any difference, since it would not have been the calculated treatment of choice. The reaming discrepancy in two further cases to the model’s calculations might not match the real treatment decision, since the model illustrates the common decision process but does not include current clinical trials. However, if the inclusion criteria for any RCT are correctly defined, the corresponding information can easily be included in the BN as a prioritized submodel to recommend this RCT whenever the eligibility criteria are met.

Our study, as outlined above, has several strengths. Based on a comprehensive search and scientific knowledge, probabilities are evidence based and approved by board-certified physicians of all professions involved in the treatment of R/M HNSCC and also include data science to consent the framework of the Bayesian network submodel. However, the validation performed with only 25 R/M HNSCC potentially eligible for PD-1/PD-L1-targeted therapy regimens can provide only proof of the principle for utilizing BN as CDSS supporting the decision-making process. As replication of our findings requires so far unavailable datasets with comprehensive characterization of all R/M HNSCC patients accrued, the only way to get a reliable validation will be the use of BN in decision-making within a prospective setting, for instance, registered research along RCTs. Indeed, we scheduled the use of an adapted version of the BN submodel in our MDTB extended with the eligibility criteria of open RCTs for R/M HNSCC.

Currently, new treatment methods are becoming more common, and likewise, more patients are receiving these therapy regimes, providing increasingly more suitable datasets. For a reliable use in a clinical context, like a tumor board, further modeling is required, as well as validation with real patients’ data to ensure a valid output. We will continue our work with the model to ensure successful validation and aim to create a larger more comprehensive model to depict a more detailed decision-making process, leading to more patient-focused and precise decisions.

## 5. Conclusions

We created a model to represent a physician’s train of thought within a tumor board regarding therapy decisions based on clinical data and additionally including pre-processed molecular information to help identify the right treatment for the right patient. As shown, the use of BNs is suitable to fulfill this task, resulting in a model that is able to depict decisions and trains of thought, resulting in a comprehensible decision. The model will be integrated into a larger decision context, so more patient-centered factors, ‘drug factors’ like their side effects and potential adverse events as well as ‘tumor factors’ related to their molecular-genetic and immunologic properties, will be respected in the future. This larger context may help to achieve improvements, resulting in more individual, patient-focused, and precise decision-making for adequately stratified or even personalized treatment of HNSCC patients.

## Figures and Tables

**Figure 1 cancers-13-05890-f001:**
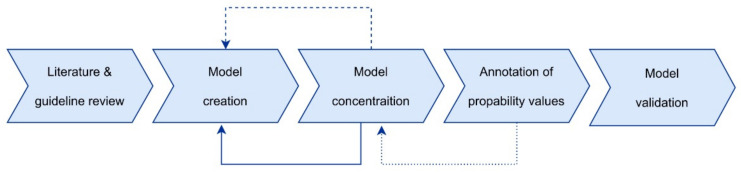
The initial literature research on HNSCC, ICB, and potential future targets of ICB led to a first graph model with more than 290 nodes. The input of regular expert meetings (depicted by the dashed arrow) and visits of the university hospital’s tumor board (depicted by the drawn trough arrow) were integrated into the model development process. After step-by-step adoption of the model (with regular meetings, depicted by the dotted line), we finalized the model for validation analysis.

**Figure 2 cancers-13-05890-f002:**
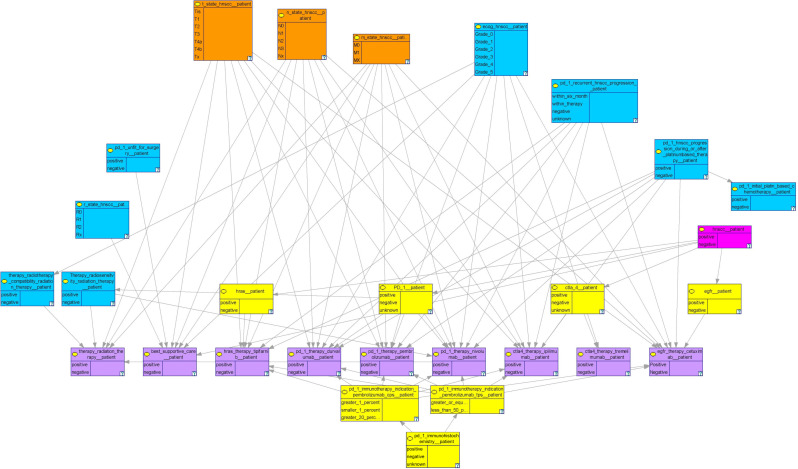
An overview of the complete model. Thematically related variables share the same color. Orange: TNM, yellow: Immunohistochemistry and genetic targets; pink: HNSCC, blue: additional conditions to match indications or treatment conditions; lilac: drugs and other therapeutic measures.

**Figure 3 cancers-13-05890-f003:**
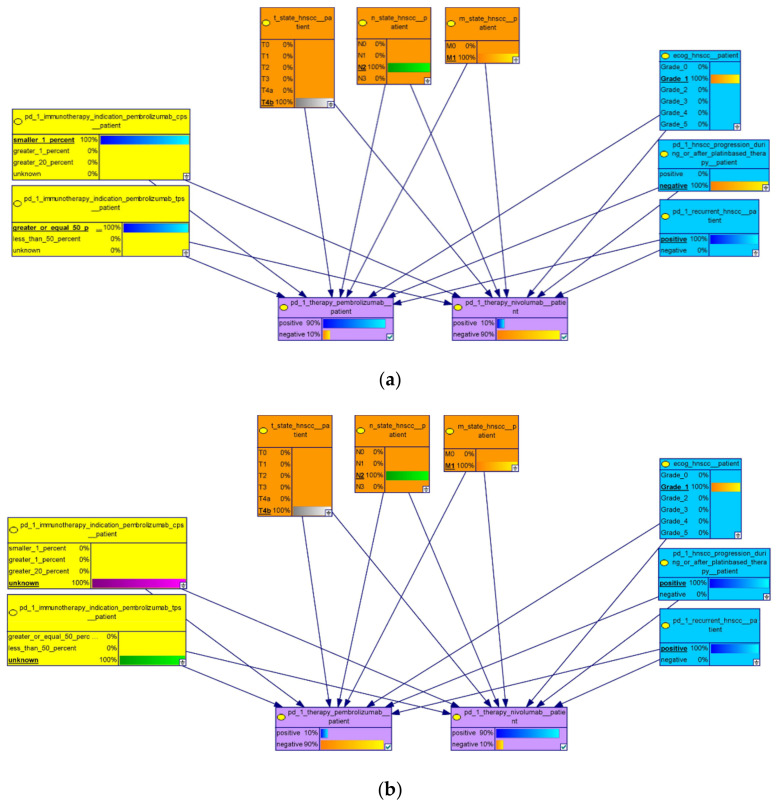
(**a**): A part of the main model, representing the essential variables, leading to a decision regarding whether Pembrolizumab or Nivolumab is a suitable choice or not. The color scheme matches the explanations above. We implemented TNM, the ECOG score and recurrence node from the blue group and PD-1, immunohistochemistry as well as CPS and TPS nodes from the yellow group. (**b**) For R/M HNSCC patients without information about the PD-L1 status and an observed progression during or after platinum-based therapy, the use of Nivolumab is recommended by the model. Because the approval of Nivolumab is independent from TPS and CPS, the probability for Nivolumab treatment increases compared to the use of Pembrolizumab (cf. CheckMate-141 and KEYNOTE-048).

**Table 1 cancers-13-05890-t001:** Proposed treatments according to the Bayesian network immunotherapy submodel for 25 HNSCC patients referred to the tumor board for “R/M HNSCC patients and immunotherapy decision”.

T	N	M	ECOG	PD	Recurrent HNSCC	CPS	TPS (%)	Actual Therapy	Treatment Decision Matches Model Result?	Calculation by Model (%)
T3	N3b	M1	1	yes	yes	n.a.	n.a.	2L Nivo	yes	Nivo 80 Pemb 20
T3	N1	M1	2	yes	yes	n.a.	n.a.	2L Nivo	yes	Nivo 85 Pemb 20
T3	N2b	M1	4	no	yes	n.a.	n.a.	BSC	yes	Nivo 10 Pemb 20
T4a	N0	M1	n.a.	no	yes	n.a.	n.a.	RCT, Nivo	no	Nivo 10 Pemb 75
T2	N3b	M0	4	no	no	11	5	PRT	yes	Nivo 10 Pemb 10
T4a	N3b	M0	2	yes	yes	n.a.	n.a.	RCT, Nivo	yes	Nivo 80 Pemb 10
T4a	N2b	M1	1	yes	yes	n.a.	n.a.	RCT, Nivo	yes	Nivo 90 Pemb 10
T2	N2	M1	2	no	yes	n.a.	n.a.	RCT, Pemb	yes	Nivo 10 Pemb 85
T2	N3b	M1	2	yes	yes	n.a.	n.a.	RCT, Pemb	no	Nivo 80 Pemb 20
T2	N2	M1	2	yes	yes	1	<1	RCT, Pemb	yes	Nivo 65 Pemb 80
Tx	N2c	M1	3	yes	yes	n.a.	n.a.	RCT, Nivo	yes	Nivo 71 Pemb 10
T4b	N3b	M0	1	yes	yes	n.a.	n.a.	RCT, Nivo	yes	Nivo 90 Pemb 10
T4a	N2c	M1	1	yes	yes	n.a.	n.a.	2L Nivo	yes	Nivo 90 Pemb 10
T3	N3	M1	0	yes	no	n.a.	n.a.	2L Nivo	yes	Nivo 80 Pemb 10
T3	N3b	M0	4	yes	no	n.a.	n.a.	BSC	yes	Nivo 10 Pemb 10
T4a	N2c	M1	1	yes	yes	n.a.	n.a.	2L Nivo	yes	Nivo 90 Pemb 10
T2	N1	M1	1	yes	no	n.a.	n.a.	2L Nivo	yes	Nivo 70 Pemb 10
T4	N2	M0	3	no	yes	1	1	RCT, Pemb	yes	Nivo 10 Pemb 65
T4a	Nx	M1	2	yes	no	15	10	2L Nivo	no	Nivo 65 Pemb 75
T3	N3b	M1	1	yes	yes	n.a.	n.a.	2L Nivo	yes	Nivo 80 Pemb 20
T4a	N0	M0	1	no	yes	n.a.	n.a.	RCT, Nivo	no	Nivo 10 Pemb 20
T4b	N2b	M1	1	yes	no	n.a.	n.a.	2L Nivo	yes	Nivo 90 Pemb 10
T3	N2c	M1	1	yes	no	n.a.	n.a.	2L Nivo	yes	Nivo 80 Pemb 10
T4b	N3b	M1	1	yes	yes	n.a.	n.a.	2L Nivo	yes	Nivo 90 Pemb 10
T4a	N2b	M1	0	no	yes	3	2	2L Pemb	yes	Nivo 10 Pemb 90

T, T category according to TNM 8th ed.; N, N category according to TNM 8th ed.; M, M category; ECOG, general health status scored according to the Eastern Collaborative Oncology Group; PD, progressing disease under or after platinum-based therapy; n.a., not available/not assessed; 1L, first-line systemic treatment; 2L, second-line systemic treatment; BSC, best supportive care PRT, palliative radio therapy; Nivo, Nivolumab; Pemb, Pembrolizumab; RCT, randomized controlled trial: PFE, cisplatin, 5-FU, Cetuximab according to the EXTREME protocol.

## Data Availability

Data is contained within the manuscript. Further information about the data may be provided on request.

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
