# Peer review of "Bayesian Networks to Support Decision-Making for Immune-Checkpoint Blockade in Recurrent/Metastatic (R/M) Head and Neck Squamous Cell Carcinoma (HNSCC)"

_cancers, 2021, doi:10.3390/cancers13235890_

Round 1

Reviewer 1 Report

An original paper proposing a bayesian model in order to select the correct chemotherapy for patients affected by HNSCC. The model reported an 84% of accuracy in predicting the actual drug used in a small cluster of patients retrospectively enrolled. Given the accuracy rate and the future perspective in chemotherapy in this area; more work will be necessary for the future to improve this model; only minor queries:

You have to specify the statistical program, the version, the maker, and its location.

page 2 line 52 you should add: "HNSCC is a highly invasive tumor, and often surgery and radiotherapy are not enough to defeat this cancer, so chemotherapy agents are necessary." and cite an article such as: doi: 10.3390/curroncol28040213. and doi: 10.3390/medicina57060563.

Author Response

We appreciate the Reviewer’s helpful comments and suggestions, which have been addressed as follows:

Reviewer’s Comments:
Reviewer 1:

Comment of reviewer:

            English language and style are fine/minor spell check required

Response to reviewer:

            We conducted a check and correction of spelling and style wherever appropriate.

Comment of reviewer:

An original paper proposing a Bayesian model in order to select the correct chemotherapy for patients affected by HNSCC. The model reported an 84% of accuracy in predicting the actual drug used in a small cluster of patients retrospectively enrolled. Given the accuracy rate and the future perspective in chemotherapy in this area, more work will be necessary for the future to improve this model; only minor queries:

Comment of reviewer:

  1. You have to specify the statistical program, the version, the maker, and its location.

Response to reviewer:

Thank you for pointing out the correctness in the description of the utilized Software. We used the modeling software GeNIe, distributed by BayesFusion Version 2.2. It is an academically free to use program originating from Marek J. Druzdzel at the University of Pittsburgh in the second version in 2000. SPSS version 24 (IBM Corp. IBM SPSS Statistics for Windows, Version 24.0, 2016. Armonk, NY: IBM Corp.) was used to assess differences between categorical variables utilizing Pearson’s Chi-square (χ2) tests odds ratios and confidence intervals.

We added Version number, distributor and location in the footnote. Also SPSS specificities were introduced in the manuscript.

Comment of reviewer:

  1. Page 2 line 52 you should add: "HNSCC is a highly invasive tumor, and often surgery and radiotherapy are not enough to defeat this cancer, so chemotherapy agents are necessary." and cite an article such as: doi: 10.3390/curroncol28040213. and doi: 10.3390/medicina57060563.

Response to reviewer:

Thank you for this valuable remark. We appreciate the clarification of HNSCC hazardousness and necessity of comprehensive treatment supported by most recent literature.

We added the suggested sentence and first suggested reference to improve the manuscript. In addition, we considered the second suggested reference (10.3390/medicina57060563) very narrow regarding the focus of our manuscript and therefore abstained from quoting the latter.

Reviewer 2 Report

In the present article, the authors design a digital patient model to assist the immunotherapy treatment in a clinical setting.  I have several reservations, my comments are appended as below:

  1. Introduction- authors should share details on prognosis details on chemo/immunotherapy treatment.
  2. In the introduction section, the authors need to describe the mechanisms of immunotherapy action more precisely. For instance, there are other mediators like TIM3, TIGIT in addition to conventional ones and discussed well in the existing literature.
  3. Reference 6,7- do these studies involves human patients? If yes, then authors should share details on the number of patients and the statistical inference.
  4. Reference 22- please elaborate.
  5. Authors seem to discuss the cofounders as a mutational burden, in addition, should also discuss other factors as diet, metabolic disorders again reviewed in the existing literature. PMID: 33076303, PMID: 33909988.
  6. It is noteworthy to create a model, but its validity in existing cohorts should be testified. I observe that authors do validate, but the sample size is less, authors try to see if other datasets are available for robustness.
  7. Authors should share a flow chart on model creation.
  8. Conclusion- authors should discuss the integration of other decision-making factors like obesity, diet, etc also noted in earlier point 5.
  9. Authors should present hypothesis figures for better understanding.

Author Response

We appreciate the Reviewer’s helpful comments and suggestions, which have been addressed as follows:

Reviewer’s Comments:

Reviewer 2:

Comment of reviewer:

            English language and style are fine/minor spell check required

Response to reviewer:

            We conducted a check and corrected spelling and style wherever appropriate.

Comment of reviewer:

  1. Introduction- authors should share details on prognosis details on chemo/immunotherapy treatment.

Response to reviewer:

We are grateful that the reviewer pointed to this significant aspect of recurrent/metastatic (R/M) HNSCC treatment and gladly elaborate on this very important principle of the manuscript. In R/M HNSCC, different treatment options are available, with immunotherapy leading to a recent adjustment of the guidelines regarding systemic therapy, first-line (1L) and second-line (2L) treatment of R/M HNSCC in particular. Nivolumab or Pembrolizumab entered stage in HNSCC treatment leading to significant improved outcome.

After platinum based therapy, an immunotherapy with Nivolumab causes a longer median overall survival with 7.5 months (95% confidence interval (CI) 5.5 to 9.1) compared to a standard chemotherapy. The standard therapy consisted of methotrexate, docetaxel or Cetuximab leading to an overall survival of 5.1 month (95% CI 4.0 to 6.0). Within all 361 patients, overall survival was significantly longer under therapy with Nivolumab, received by 231 patients.
Overall survival under Pembrolizumab therapy was also prolonged in KEYNOTE-048. A comparison between the treatment with Pembrolizumab alone (301 patients), Pembrolizumab with 5-floururacil (5-fu) (281 patients) and Cetuximab with 5-fu (300 patients), showed prolonged survival of any treatment combination with Pembrolizumab. Combined with 5-fu, Pembrolizumab showed a median overall survival of 13.0 month versus 10.7 month (Hazard ratio (HR) 0.77 [95% CI 0.63–0.93) in the Cetuximab plus 5-fu group. Pembrolizumab alone showed a median overall survival of 11.6 month, still being superior (0.85 HR 0.71–1.03).
Dividing 495 patients into even groups, the KEYNOTE-040 trial compared a treatment with Pembrolizumab versus the investigator’s choice of methotrexate, docetaxel or Cetuximab. The median overall survival in the group with Pembrolizumab was 8.4 month, compared to 6.9 month in the common chemotherapy group (hazard ratio 0.80, 0.65-0.98).

Therefore, we specified the introduction section regarding these breaking-through studies by providing details of the study protocols and outcomes:

Ferris et al. reported a significantly prolonged OS for treatment with Nivolumab compared to a standard therapy of methotrexate, docetaxel or Cetuximab. The 231 patients receiving Nivolumab showed a median OS of 7.5 months (95% confidence interval (CI) 5.5 to 9.1) versus 5.1 months (95% CI 4.0 to 6.0) of the 130 patients receiving standard therapy [8]. Likewise, Pembrolizumab achieved a median OS of 8.4 months versus 6.9 months in the common treatment group (hazard ratio (HR) 0.8, 0.65-0.98) [9].

Comment of reviewer:

  1. In the introduction section, the authors need to describe the mechanisms of immunotherapy action more precisely. For instance, there are other mediators like TIM3, TIGIT in addition to conventional ones and discussed well in the existing literature.

Response to reviewer:

This is a valuable addition and optimization of to our line of reasoning, so we are grateful for this wonderful suggestion.

This is a valuable addition and optimization of to our line of reasoning, so we are grateful for this wonderful suggestion.

HNSCC often exhibit a low count of lymphocytes including natural killer cells, causing an immunosuppressive state. Immune checkpoint blockage (ICB) can inhibit pathways involved in the tumor’s immune escape mechanisms advancing tumor treatment.

Altered (reduced) expression of human leukocyte antigens (HLA) and/or reduced antigen processing and presentation of peptides derived from tumor-associated antigens (TAA) can reduce the recognition of HNSCC by T cells via their T cell receptor (TCR), and reduced HLA expression is a long known characteristic of HNSCC, causing immune evasion. A complete loss of HLA expression, however, could cause the activation of Natural Killer Cells, as NK cells are attacking HNSCC with complete loss of HLA expression, loss of HLA-C in particular.

T cells can bind with their TCR to peptides derived from TAA in HLA together with either CD4 or CD8 and form an immunologic synapse providing the first activating signal. The second (activating) signal is derived from interaction of CD28 on T cells with B7 proteins (B7.1=CD80 on monocytes, B7.2=CD86 on B cells, both expressed on dendritic cells, DC) on the antigen presenting cell (APC). However, CTLA-4 is a B7 ligand potentially involved and rather dampening immune responses as we outlined in our manuscript However, additional immune checkpoints could be involved in immune escape of HNSCC.

Accordingly, we provided more detail on the above-mentioned mechanisms with pertinent references in the manuscript:

Like PD-1, CTLA-4 balances the immune response, usually stopping an exaggerated immune response, so the immune system does not damage the host. Cancers abuse this mechanism by secretion of tumor growth factor β (TGF-β), causing CTLA-4 expression on T cells and expression of B-7, a ligand of CTLA-4. This combination leads to an ex-haustion of T cells, allowing the cancer to escape immune response. Another way to by-pass the immune system is achieved by the T cell immunoglobulin mucin-3 (TIM-3), which is commonly expressed by cells of the immune system. After binding its ligand, galectin-9, TIM-3 induces apoptosis and exhaustion in NK cells and T cells [19]. Fur-thermore, it promotes the expansion of bone marrow derived suppressor cells (MDSCs), preventing adequate immune response. Anti-TIM-3 antibodies lead to decreased counts of MDSCs and an increased T cell activity in a HNSCC mouse model [20].

Comment of reviewer:

  1. Reference 6,7- do these studies involves human patients? If yes, then authors should share details on the number of patients and the statistical inference.

Response to reviewer:

We thank the reviewer to ask for clarification regarding this issue that alludes to the reviewers comment number 1.

Indeed, our references number six and seven involve patients. Review number six is the work of Ferris et al. including 361 patients within his study and dividing them into two groups of 231 patients receiving Nivolumab and 130 patients receiving standard therapy as stated above. While comparing Nivolumab with standard therapy, the median overall survival was significantly longer within the Nivolumab group 7.5 months (95% CI, 5.5 to 9.1) versus 5.1 months (95% CI, 4.0 to 6.0) as well as overall survival (HR for death, 0.70; 97.73% CI, 0.51 to 0.96; P = 0.01).
Cohen’s work is our seventh reference. He included 495 patients in his research, splitting the group in half and assigning 247 patients randomly to Pembrolizumab and 248 randomly to standard care, as mentioned above. The median overall survival in the group receiving Pembrolizumab was 8.4 month (95% CI 6.4–9.4), compared to 6.9 month (95% CI 5.9 – 8.0) in the common chemotherapy group (hazard ratio 0.80, 0.65-0.98 nominal p=0.0161).

We adjusted the details in the manuscript as stated above in response 1.   

Comment of reviewer:

  1. Reference 22- please elaborate.

Response to reviewer:

      Reference number 22 is the work of Mandal et al. He states, that HNSCC benefits massively from its immune escape mechanisms. These mechanisms yield great potential of future pathway blockage. CTLA-4 and TIGIT as examples described earlier, as well as other potential targets, preventing T cell exhaustion and thus preserving their antitumor effects. Besides exhausted T cells, inactive NK cells are the result of the tumors evasion mechanisms. Complex interactions of TIGIT, TIM-3 and the lymphocyte activation gene (LAG-3) are linked to the inhibition of NK cells, therefore altering their immunologic abilities, resulting in a favorable immunologic milieu for HNSCC. Blockage of these factors might be advantageous HNSCC, since it shows high levels of NK cells.

We elaborated on this issue by augmentation of the mentioned paragraph within the manuscript as follows:

Although therapies for HNSCC treatment have been approved by the FDA, HNSCC still bears many potential options for targeted therapy [26]. Besides mentioned antigens, other mediators of immune reaction may be expressed by the tumor cells evading the immune system. E.g. the lymphocyte activation gene (LAG-3) is linked to the inhibition of NK cells, altering their immunologic abilities, resulting in a favorable immunologic milieu for tumor cells [26]. NK cell functions are also altered by the T cell immunoglobulin and immune receptor tyrosine based inhibitor motif (TIGIT), which causes a reduction of pro-inflammatory cytokines and reduced immunologic response [20,27].

Comment of reviewer:

  1. Authors seem to discuss the cofounders as a mutational burden, in addition, should also discuss other factors as diet, metabolic disorders again reviewed in the existing literature. PMID: 33076303, PMID: 33909988.

Response to reviewer:

This is an important issue in cancer treatment and of increasing interest of researchers in HNSCC. Different alterable and unalterable factors affect tumor genesis and treatment. HNSCC manipulates the immune system and thus escaping fundamental immune response. Several factors can influence our immune system toward either better or even worse response against HNSCC or efficacy of ICB treatment.
Ageing is associated with a decline in immune functions, a misbalance of immunogenic factors. Older Adults suffer from side effects of a comparable level like younger adults. However, antigens like TIMS-3 or PD-1 are upregulated, thus causing a decreased immune response. While the total amount of naïve T cells decreases, possibly due to reduced thymus size, the amount of regulatory T cells (Treg), inducing an immune suppression, was increased.

Diet itself and the affected microbiome can influence the efficacy of ICB. A balanced and healthy diet is essential for building and maintaining a robust immune system. Specific diets can alter cytokine levels measured in the saliva. A junk food associated diet was shown to increase levels of IL-17. LAG-3 positive T cells are associated with secretion of IL-17, thus patients with IL-17 overexpression might benefit from treatment with anti-LAG-3 Antibodies, and likewise children with a Mediterranean diet present high cytokine IL-10 levels. In melanoma patients, the treatment with Nivolumab showed higher response rates in patients with higher IL-10 levels. However, IL-10 inhibits proliferative T cell responses, by reducing the secretion of pro-inflammatory cytokines. Diet also affects the human gut and oral microbiome. Specific microorganism can have an effect on ICB. Experimental data suggests, that B. fragilis enhances anti-CTLA-4 efficacy, due to cross reactivity between bacterial and cancer cell antigen. Melanoma patients with Bacteroides phylum in stool samples, receiving anti-CTLA-4 antibodies were less likely to suffer from treatment-induced colitis. Oral squamous cell carcinoma (OSCC) may increase with periodontitis. Inflammation via IL-1 and TNF-α is linked to microbial triggered cancers, supporting the strong interplay between HNSCC and the host’s immune system.

The precise impact of bacteria on therapy regimes, tumor entities or side effects is complicated by intermeshed mechanism, different bacteria species and their diverse attributes, and until now could not reliably incorporated into a model aiming on supporting clinical decision-making for the probably best guideline-conform treatment regimen.

To address the abovementioned issues, we inserted a separate paragraph in the manuscript:

Despite many genetic alterations, various other factors can influence the tumor gen-esis and the therapeutic process. The immune system shifts towards a misbalance in older patients [29]. The thymus releases a greater amount of regulatory T cells, causing a weakened immune response. Simultaneously antigens like TIM-3 or PD-1 are upregulated on T cell surfaces, leading again to reduced immune response [30,31]. Although the immune response toward the cancer may be reduced, side effects during ICB therapy occur on comparable levels in young and old adults [30].

Comment of reviewer:

  1. Conclusion- authors should discuss the integration of other decision-making factors like obesity, diet, etc also noted in earlier point 5.

Response to reviewer:

With regard to the previous comment of the reviewer, we appreciate this issue to be included in the discussion of the manuscript.

Nivolumab and Pembrolizumab are the approved ICB for HNSCC; they come with known and proved indications that we depicted in our model. Therefore, these parameters are regularly recorded during the process of diagnostics and treatment decision leading to verifiable data and information integration.

Research and literature offer hints of interaction mechanisms of the above-mentioned influence parameters on HNSCC development and treatment, yet data is inconsistent today regarding the exact dependencies. This prevents integration of these parameters in our model. Still, we attentively observe research in this field and plan to integrate these parameters in an extended model if clear dependencies are described.

Until today, the addressed factors like diet or state of the microbiome with its colonization of microbes are not documented (regularly) in our clinical routine, as well as in pertinent studies leading to the approval of either Pembrolizumab or Nivolumab. Consequently, the impact on parameters like overall survival remains yet unclear. Missing documentation hinders integration of factors into our model. However, since the microbiome and thus our diet can alter the body’s reaction towards ICB we are convinced that there is need to extend the model in the future, when more knowledge is adapted. Our model could potentially aid these decisions, supporting the physician to facilitate more information and variables. 

We clarified this issue in our manuscript in the discussion section as follows:

We did not include factors like age, diet or specific colonization of either gut or oral microbiome into our model. Measured data and potential (treatment-) consequences are not well established, and consequently not measured. A wider spectrum of different nodes could depict a more exact model of the patient.

Comment of reviewer:

  1. It is noteworthy to create a model, but its validity in existing cohorts should be testified. I observe that authors do validate, but the sample size is less, authors try to see if other datasets are available for robustness.

Response to reviewer:

Our aim was to create a model with the ability to verifiable differentiate between potential ICB Pembrolizumab and Nivolumab. We regard our analysis a proof-of-concept study by the retrospective analysis of the model. The results suggest the significant viability of the model. A greater number of included patients could aid the graph in terms of robustness nevertheless, but our current objective was to prove our concept of a Bayesian network for treatment decision support in HNSCC. The used software and our attempt to create the model are working. In addition, study cohorts with the same pathological survey and detailed patient data are not available open source to our knowledge. We are committed to conduct prospective trials to further validate the model in the future. Yet, this issue is way beyond the status of our research and therefore not within the scope of presented study and this manuscript.

Comment of reviewer:

  1. Authors should share a flow chart on model creation.

Response to reviewer:

We thank the reviewer for this valuable advice of improving the manuscript. We explained the steps of the model creation in the methods section. Still, we are well aware that a figurative demonstration of the process helps to understand the methods for the reader. Consequently, we included the following figure and subscription within the manuscript.

Figure 1: Initial literature research on HNSCC, ICB and potential future targets of ICB led to a first graph model with more than 290 nodes. Input of regular expert meetings (depicted by the dashed arrow) and visits of the university hospital’s tumor board (depicted by the drawn trough arrow) were integrated the model development process. After step-by-step adopting the model (with regular meeting, depicted by the dotted line), we finalized the model for conducting a validation analysis.

Comment of reviewer:

  1. Authors should present hypothesis figures for better understanding.

Response to reviewer:

According to the previous comment, we may provide hypothesis figures of the presented work for visualization and comprehensibility:

            Figure 2:

Our idea is to support the physician in the decision making process: The tumor board discussion is the center point in this process, since evidence from clinical trials and molecular data before are discussed in a comprehensible before decisions are consented. Input from literature and guidelines were integrated into our model, concentrating it to our final version. After setting the probability values as priors, we validated the graph with our cohort of 25 patients.

We did not include this hypothesis figure in our manuscript, because we believe our flow chart about the model creation process gives sufficient aid to comprehend our aims and scopes. Thereby, we avoid repetition. 

Round 2

Reviewer 2 Report

I congratulate the authors for providing the modifications, with that, the manuscript is closer to publication. I, however, suggest taking note of the following minor concern.

  1. Point 6,7- I am partly satisfied with the response provided. I suggest adding limitations of this study in the discussions section.

Author Response

Comment of reviewer:

            English language and style are fine/minor spell check required

Response to reviewer:

            We conducted another check and corrected spelling and style wherever appropriate.

Comment of reviewer:

  1. Point 6,7- I am partly satisfied with the response provided. I suggest adding limitations of this study in the discussions section.

Response to reviewer:

We are grateful that the reviewer supports the fundamental ideas and aims of our work. By adding the limitations of our model to our manuscript, we share transparency about the work and the presented results. We hope to foster a more comprehensible manuscript. Therefore, we elaborated on the two remaining suggestions for improvement:

R1 point 6. Conclusion- authors should discuss the integration of other decision-making factors like obesity, diet, etc also noted in earlier point 5.

We integrated the following section in the previous version of the manuscript (R1):

We did not include factors like age, diet or specific colonization of either gut or oral microbiome into our model. Measured data and potential (treatment-) consequences are not well established, and consequently not measured. A wider spectrum of different nodes could depict a more exact model of the patient.

Still, we understand that a broader discussion of this issue improves the manuscript. Therefore, we further elaborated the above mentioned paragraph in the discussion section as follows in the secondly revised manuscript:

The presented functioning model is a submodel of our main model that was concentrated by current clinical and therapeutically relevance. Therefore, we avoided to include any further information for the reason of traceability of the model.

This includes potential influencing parameters like the patient’s diet, microbiome or other patient-related factors. Therefore, the model can calculate the probability of treatment with Pembrolizumab or Nivolumab based on the approvals, but not on other potentially influencing parameters.

R1 point 7. It is noteworthy to create a model, but its validity in existing cohorts should be testified. I observe that authors do validate, but the sample size is less, authors try to see if other datasets are available for robustness.

We acknowledge that patient numbers provided are limited as discussed in Review 1 point 7. This aspect is pointed out in our manuscript.

We put more emphasize on that issue by adding the following section in the manuscript:

We are aware that the evaluation of the model with 25 patient cases implies limited expressiveness. The presented analysis was conducted for the purpose of proof of concept. The results of our validation study are significant in spite of the fact that only 25 patients were analyzed. In addition, study cohorts with the same pathological survey and detailed patient data are not available open source to our knowledge.

To round up the topic, we included a last-but-one paragraph of strengths and weaknesses of the presented work:

Our study, as outlined above, has several strength. Based on a comprehensive search and scientific knowledge, probabilities could be set evidence based and approved by board-certified physicians of all professions involved in treatment of R/M HNSCC and data science as well to consent the framework of the Bayesian Network submodel. However, the validation performed with the only available 25 R/M HNSCC potentially eligible for PD-1/PD-L1-targeted therapy regimens can provide only a proof of principle for utilizing BN as CDSS supporting the decision-making process. As replication of our findings requires so far unavailable datasets with comprehensive characterization of all R/M HNSCC patients accrued, and therefore the only way to get a reliable validation will be the use of BN in decision-making within prospective setting, for instance registered research along RCT. Indeed, we scheduled the use of an adapted version of the BN submodel in our MDTB extended with eligibility criteria of open RCTs for R/M HNSCC.

Round 3

Reviewer 2 Report

ALl my concerns are now addressed. I suggest accepting this manuscript.